# Phenotypic Plasticity in Bud Fruitfulness Expressed in Two Distinct Wine Grape Cultivars Grown under Three Different Pedoclimatic Conditions

**Elisabetta Nicolosi** [1],[†] , **Angelo Sicilia** [1],[†] , **Filippo Ferlito** [2],[*] , **Antonello Bonfante** [3] , **Eugenia Monaco** [3] and **Angela Roberta Lo Piero** [1]





1   Department of Agriculture, Food and Environment, University of Catania, Via Valdisavoia 5,
    95123 Catania, Italy
2   Council for Agricultural Research and Economics, Research Centre for Olive, Fruit and Citrus Crops,
    Corso Savoia 190, 95024 Acireale, Italy
3   Institute for Mediterranean Agricultural and Forest Systems—CNR-ISAFOM, National Research Council,
    Piazzale Enrico Fermi 1, 80055 Portici, Italy
*   Correspondence: filippo.ferlito@crea.gov.it; Tel.: +39-095-7653106
†   These authors equally contribute to the work.

**Abstract:** The effects of interactions between the genotype and environmental conditions are ex-
pressed in the phenotype. Comparing the performances of genotypes under the same range of
environmental conditions allows for relative measurements to be made of the different levels of plas-
ticity among those genotypes. The objective of this research was to evaluate the different responses
of two wine grape cultivars, native Aglianico and international Cabernet Sauvignon, under different
pedoclimatic conditions in terms of the functional traits that govern grapevine bud fruitfulness,
vegetative growth, and yield development. The study was conducted over two consecutive seasons
(2020 and 2021), in six commercial *Vitis vinifera* L. vineyards, located in three distinct viticultural
regions of central-southern Italy (Molise, Campania, and Sicily). In each experimental vineyard, the
bud fruitfulness, number of leaves, total leaf area per vine, midday vine water status, and fruit yield
were measured. The obtained results showed that bud fruitfulness was higher for Aglianico than for
Cabernet Sauvignon in each experimental site, while the variability of plant responses between the
vineyards was lower for Cabernet Sauvignon cultivar compared to those for Aglianico. The genetic
× environmental interactions were expressed predominantly during berry ripening stages, while
plasticity was generally greater in Aglianico than in Cabernet Sauvignon.

**Keywords:** climate change; Aglianico; Cabernet Sauvignon; morphology; leaf area; leaf water
potential; canopy; drought stress

## 1. Introduction

The viticultural sector is one of the most important high-income agricultural sectors
that is affected by climate change (CC) [1]. In the last decade, the understanding of
vine cultivars' adaptability and their phenotypic plasticity is becoming fundamental to
supporting the resilience of local viticultural systems and their wines' typicity. The latter is
a direct consequence of interactions between genotypic (G) and environmental (E) factors,
which are modulated by human activities, as described by [2], and summarized by the
terroir concept of OIV (*terroir* is a concept that refers to an area in which the collective
knowledge of interactions between the identifiable physical and biological environment and
applied viticultural and enological practices develops, providing distinctive characteristics
for the products originating from this area). *Terroir* includes specific soil, topography,
climate, landscape characteristics, and biodiversity feature (OIV resolution 333/2010).

Thus, plant growth and development (e.g., the number and size of a plant's organs) are
the results of the interactions between G and E drivers, affecting phenotypic expression [3,4].

G × E interactions affect the agronomic performance of a specific genotype, under different environments [5]. The plant attribute, 'phenotypic plasticity', is a measure of the capacity of a genotype to express different phenotypes in response to different environmental conditions [6,7]. Plant phenotype is sensitive to both spatial and temporal variability in environmental conditions (climatic and soil characteristics) and to cultural practices [8], which drive plant adaptation. Then, if the environmental drivers are well identified and thresholds are defined in terms of effects on plant responses, it is possible to evaluate the impacts of both spatial relocation (to a different environment) and climate change (a temporal change in climate) for a specific genotype on the plant's ecology (annual cycle, dormancy etc.) [9], and thus, on agronomic performance (bud fruitfulness, growth, flowering, nutrition, yield) [10].

The grapevine (*Vitis vinifera* L.) has been the subject of a wide range of studies which focused on the G x E interaction [11,12]. The grape characteristics of a specific grape cultivar can change drastically, as the environmental conditions and management of the grape variety change [13]. Moreover, under the same climate and agricultural management practices, two different soils can lead to very different vine responses in terms of berry characteristics. This means that plant plasticity can be expressed and evaluated in consideration of different pedoclimatic conditions.

Moreover, according to the terroir concept, the wine produced in a specific region is unique, and is not reproducible in other places, inasmuch as it is strongly dependent on natural and human factors involved in wine production that cannot be transferable in other contexts (pedoclimatic conditions, biological material, cultural, and socio-economic factors). For this reason, wines in France have the names of their regions of origin, differently from other countries in which the name is based on the variety (e.g., Cabernet Sauvignon, Aglianico, etc.). Meanwhile, in a culturally and climatically heterogeneous nation, such as Italy, it is considered that few of the so-called 'traditional cultivars' consistently show their best traits, except in their own dedicated environment [14]. This suggests that greater variability in fruit quality, or irregular growth and ripening, will occur if the cultivar is grown in another area [15,16]. On the other hand, some allochthonous and less plastic cultivars, such as Cabernet Sauvignon and Syrah, seem to perform well and more similarly in a wider range of regions [17].

Among the main environmental stresses that can affect growth, there are droughts, high temperatures, excess solar radiation, ultraviolet radiation, heavy metals, and salts. For perennial fruit crops, adaptation strategies to drought can be physiological, with the general effect being increasing water use efficiency [18–21]. For grapevines, water status has a strong physiological impact as the primary driver of plant physiology and grape ripening [22], and is the reason for which it is considered a key factor in the variability of plant response in different *terroirs* [23]. Grapevine water status and grape composition depend on the variability of soil attributes [23–26], independently of irrigation [27,28]. The interactions between soil–plant and atmosphere affect root hydraulics [29–31], leaf water potential, and stomatal conductance [32,33]. Grapevines have developed a range of physiological and morphological mechanisms that help sustain growth and productivity under water-limiting conditions [34]. Morphological mechanisms include adaptations such as changes in leaf area [35], in root/shoot ratio [36], and in xylem vessel size and number, and thus, in conductance [37]. In general, the earliest and highly evident response to a new environmental limitation is a change in plant vigour. Vigour is a measure of the plant's capacity to assimilate, store, and use carbohydrates; it is reflected in shoot length and canopy area growth [38,39]. Plants that grow under stress conditions may exhibit less root and canopy development [40]. For grapevines, in particular the isohydric cultivars such as Grenache, growing under stress conditions tends to exhibit conservative patterns of leaf anatomy, which seem designed to avoid the production of structures that are too expensive to be sustained [41]. Hirose and Werger [42] suggest that nitrogen (N) varies with light availability and photosynthetic capacity, with leaf N content generally higher under high light conditions. However, for the grapevine, Guilpard et al. [43] reported that

decreases in leaf N and moderate water stress during flowering in the previous season can reduce bud fertility in the subsequent season by 36 to 40%, although this response is dependent on both G and E. While grapevines show different performances under different agronomic practices [44], the plasticity of reproductive behaviour is generally high under conditions of environmental stress, and this plasticity is mediated via changes in source/sink competition [45]. Fruit yield depends on the ratio of vegetative to reproductive growth, which is affected by the trophic competition between growth sinks and the overall source–sink balance of the plant [46]. Moreover, the biennial nature of a grapevine's floral phenology plays a key role in floral induction, initiation, and differentiation [47]. The initial stages of floral induction, bud initiation, and bud differentiation are completed in the first season, with the final stages of flower formation and development occurring in the second season [48]. The early stage is thus critical, because flower production depends on growth conditions from the previous season's floral period [43]. A number of factors affect the balance between vegetative and reproductive growth; these include both endogenous and exogenous factors. Exogenous factors are linked to the weather (especially to temperature and light) and also to the soil (especially to water and mineral status). This study is based on the hypothesis that the level of plasticity in different grape cultivars, and hence their morphological and physiological behaviours under adverse environmental conditions, may provide an indicator of that cultivar's ability to adapt in the context of climate change. Thus, how growth rate reduces under increasingly limiting conditions of drought, high temperatures, and solar radiation, has been monitored. Alternatively, when the environment is not limiting, extra biomass is accumulated that is available to be partitioned into increased canopy growth, with increased shoot length, leaf number and leaf size, with more lateral shoots and higher yields. The objective of this article was to show, for two distinct wine grape cultivars, Aglianico (AGL, national cv) and Cabernet Sauvignon (CAB, international cv), how the plant adapts to different environmental conditions that govern bud fruitfulness, vegetative growth, vine water status, and yield. The main results of this study should be of immediate practical use to grapevine growers, but also to ecologists who specialise in crop climate change-related responses, since the results should provide insights into the complex system that connects the interactions between G and E in determining phenotype.

## 2. Materials and Methods

### 2.1. Site Descriptions, Plant Material, and Trial Design

The research was conducted over two seasons (2020 and 2021), in six commercial vineyards that were located in three central and southern regions of Italy: Molise (MOL), Campania (CAM), and Sicily (SIC). The vineyard geo-references and main climate characteristics are detailed in Table 1. For each region, two vineyards were chosen. Each vineyard grew two black (*Vitis vinifera* L.) wine grape cultivars: 'Aglianico' (AGL), a traditional Italian cultivar, grown mainly in southern Italy in Campania and Basilicata, and the French cultivar 'Cabernet Sauvignon' (CAB), now considered ubiquitous and grown widely around the world. The vine responses were analysed on the basis of 17 different vegetative, physiological, reproductive, and nutritional variables, which were recorded at intervals during the overlapping vegetative and reproductive cycles, and at harvest. These variables are related to a number of distinct growth processes, including floral induction (bud fruitfulness), organogenesis (shoot and bunch initiation), morphogenesis (leaf number, leaf area), biomass production and biomass allocation, total leaf area (TLA), and fruit yield.

**Table 1.** Geolocations and data relating to chill hours (from 1 October to 28 February), growing degree days calculated from local meteorological data for the years 2020–2021, soil moisture, and temperature (averaged along soil profiles), from 1 April to 31 October, in the three locations: Molise (MOL), Campania (CAM), and Sicily (SIC).

| Site | Latitude | Longitude | Elevation [m] | Chill Hours (Hours < +7 °C) | Growing Degree Days | Soil Moisture [%] | Soil Temperature |
|---|---|---|---|---|---|---|---|
| | | | | | | | [°C] |
| MOL (San Biase, CB) | 41°72′ N | 14°57′ E | 600 | 1560 | 1650 | 55 | 12.8 |
| CAM (Galluccio, CE) | 41°33′ N | 13° 89′ E | 125 | 840 | 2112 | 41 | 17.8 |
| SIC (Zafferana Etnea, CT) | 37°41′ N | 15° 7′ E | 720 | 580 | 2045 | 15 | 16.5 |

All vines were grafted onto 140 Ruggieri rootstocks, and were about ten years old, planted between 2008 and 2010. In each vineyard, vines were planted in north-south rows, on gentle slopes in MOL and CAM, or on steeper slopes in SIC. In MOL, both cultivars were planted at a spacing of 1.20 m (in-row) × 2.90 m (between-row); both were simple Guyot trained, with a formation height of 60 cm. In CAM, the AGL vines were planted at a spacing of 1.50 m (in-row) × 2.90 m (between-row), while CAB was planted at a spacing of 1.0 m (in-row) × 2.70 m (between-row); both were simple Guyot trained, with a formation height of 60 cm. In SIC, the AGL vines were planted at a spacing of 1.10 m (in-row) × 1.10 m (between-row), while CAB was planted at a spacing of 1.10 m (in-row) × 1.30 m (between-row); both were bush trained at 0.5 m, with two to six main branches; each branch was spur-pruned to one spur, with two buds per spur.

After flowering all of the vines were standardised (based on bud load that was derived from the winter pruning) by removing by hand all shoots derived from the *bourillon* (first proximal small bud), the crown, and any adventitious buds, to retain only those shoots that derived from the main bud [9]. The shoots were vertically positioned and aligned in the row direction.

Only during the first year, shoots were hedged in mid-summer to avoid late fungal diseases, in both vineyards in Molise and Campania. None of the six vineyards was irrigated, and all vineyards were mechanically tilled between rows. The trial involved a completely randomised design, with three independent plots of five rows, each containing thirty vines. All measurements were made on seven 'index' vines per 21-vine block. For AGL, three sites and two years were considered; meanwhile, the Campania site was missed in the first season for CAB.

### 2.2. Pedoclimatic Conditions

Daily temperature and rainfall data were provided by a recording weather station located in each vineyard within the canopy. For each year, the daily minimum, mean, and maximum air temperature data, in addition to rainfall, reference evapotranspiration, and vapour pressure deficit data, were collected. The soils of selected experimental sites were very different: a deep clay soil in MOL (Vertisol), a deep clay loam volcanic soil in CAM sites (Andosol), and a shallow volcanic sandy soil in the SIC site (Andosol). The physical and chemical characteristics of the soils are reported in Table 2.

**Table 2.** Main physical and chemical parameters for soil in each study area: Molise (MOL), Campania (CAM), and Sicily (SIC).

| Parameters | Experimental Site | | |
| --- | --- | --- | --- |
| | MOL (San Biase, CB) | CAM (Galluccio, CE) | SIC (Zafferana Etnea, CT) |
| pH | 8.3 | 7.1 | 7.0 |
| EC (dS/m) | 0.135 | 0.047 | 0.034 |
| Sand (coarse) (g/kg) | 56.4 | 172.5 | 439.9 |
| Sand (fine) (g/kg) | 234.2 | 297.7 | 478.9 |
| Silt (g/kg) | 195.0 | 215.9 | 68.4 |
| Clay (g/kg) | 514.2 | 313.9 | 12.7 |
| Organic carbon (g/kg) | 10.8 | 3.7 | 25.0 |
| Organic matter (g/kg) | 18.6 | 6.4 | 43.2 |
| Nitrogen (g/kg) | 1.3 | 0.4 | 2.3 |
| C/N ratio | 7.6 | 7.0 | 10.7 |
| Assimilable phosphorus ($P_2O_5$) (mg/kg) | 25.9 | 42.5 | 23.9 |
| CSC (meq/100 g) | 31.0 | 17.4 | 17.3 |
| Exchangeable Ca (meq/100 g) | 25.7 | 3.7 | 9.8 |
| Exchangeable Mg (meq/100 g) | 3.4 | 1.8 | 0.7 |
| Exchangeable Na (meq/100 g) | 0.3 | 0.1 | 0.1 |
| Exchangeable K (meq/100 g) | 1.4 | 1.5 | 0.6 |
| Exchangeable Fe (mg/kg) | 14.8 | 49.0 | 61.5 |
| Exchangeable Cu (mg/kg) | 17.0 | 4.5 | 4.5 |
| Exchangeable Zn (mg/kg) | 2.6 | 5.9 | 0.9 |
| Exchangeable Mn (mg/kg) | 6.9 | 56.6 | 0.5 |

## 2.3. Cultivar + Environmental Combinations

There were six cultivars + terroir combinations, which were the following: (1) Aglianico-Molise (AGL-MOL), (2) Cabernet Sauvignon-Molise (CAB-MOL), (3) Aglianico-Campania (AGL-CAM), (4) Cabernet Sauvignon-Campania (CAB-CAM), (5) Aglianico-Sicilia (AGL-SIC), and (6) Cabernet Sauvignon-Campania (CAB-SIC).

## 2.4. Bud Fruitfulness and Vegetative Behaviour

In each growing season, for each cultivar x site combination, when the shoot length was approximately 40 cm (Biologische Bundesanstalt, Bundessortenamt and CHemical industry—BBCH: inflorescence 57) [49], the number of shoots and bunches deriving from nodes on spurs, *bourillon* (the bud between the crown bud and the first main bud on the spur), the crown (the bud between the old wood and the *bourillon*), and latent buds were determined. The number of blind buds on the main nodes was also recorded. After this stage, the shoots derived from *bourillon*, crown, and latent buds were hand-pruned, so that only the main shoots derived from the buds remaining after the winter pruning were retained. The potential and observed bud fertility were assessed on these shoots. Potential bud fertility was calculated as the ratio of the number (*n*) of bunches on main shoots/the number (*n*) of main shoots. The observed bud fertility was calculated as the ratio of the number (*n*) of bunches on each shoot/the number (*n*) of buds retained on the spurs [14].

## 2.5. Vegetative Behaviour and Vine Water Status Measurements

In order to record the vegetative behaviour, observations were made (a) at the principal growth stage 6: flowering (BBCH69 end of flowering); (b) at the principal growth stage 7: development of fruits (BBCH75 berries pea-sized); at (c) the principal growth stage 8: ripening of berries (BBCH85 softening of berries), in both years and on the same day of

the year; and (d) at principal growth stage 5: inflorescence emergence (BBCH57 flowers separating), during the second season.

Two leaves per vine were collected for water potential measurement in the field (see below), one from the main shoot and one from a lateral shoot. These leaves (from main and lateral shoots) were then transferred to the laboratory for leaf area (LA) measurement using an area meter (model LI-3100; Licor, Inc., Lincoln, Nebraska). Moreover, the numbers of main shoot leaves and lateral shoot leaves, in addition to the length (cm) of the main and the lateral shoots, were measured [17,18]. The total leaf area (TLA) per shoot was then calculated as the sum of the main and lateral leaf areas of the shoots. The total leaf area per shoot and the number of shoots per vine were used to estimate the total leaf area per vine (TLA/vine). The ratio of TLA and the projected canopy area of each vine were used to estimate leaf area index (LAI) at each sampling time.

Midday leaf water potential (between 12:00 and 13:30 h) was measured using a Schöelander pressure chamber (Soil Moisture Equipment Corp., Sta. Barbara, CA, USA), according to [50]. Measurements were made at flower separation (only in the second year), at the end of flowering, when berries were pea-sized, at berry softening (*vèraison*). For each cultivar × *terroir* combination and time of measurement, these data were collected from 21 fully-exposed primary leaves, and 21 lateral leaves.

### 2.6. Crop Yield and Berry Characteristics

In both years, the bunches were harvested at maturity in early October. For yield assessment, all bunches per vine and shoot were counted and weighed, and the total fresh weight yields per hectare were then calculated. Two bunches from each index vine (42 bunches) were randomly selected and dissected, in order to determine bunch weight, berry number, and mean berry weight.

### 2.7. Statistical Analyses

Analysis of variance (ANOVA) was performed using Jamovi 2.0.0 statistical software (The jamovi project, 2021). One-way analysis of variance (ANOVA) was carried out on the differences among the canopy treatments. A post hoc analysis based on the Tukey HSD test (Tukey honestly significant difference) was carried out at significance levels (*p*-values) of 0.05, 0.01, and 0.001. Bud fruitfulness data were used for the Pearson correlation analysis; all parameters collected were computed using the psych package [51] implemented in R 4.0.3 statistical software. The $\chi^2$ test of independence was computed with R 4.0.3 statistical software (R Core Team, 2020) [52]. The comparison among the three locations was carried out through principal component analysis (PCA) of the average data using the PAST, PAleontological STatistics software package, 2011 [53], in order to summarise the specific responses of the two cultivars graphically, and to highlight any similarities or dissimilarities.

## 3. Results

### 3.1. Climatic Conditions

Monthly maximum, average, and minimum temperatures, as well as rainfall and reference evapotranspiration values, are reported in Figure 1. In Molise and Campania, the climate is characterised by cold winters and mild summers. Moreover, the rainfall in these areas is fairly uniform throughout the year, with significant amounts occurring in late spring and summer. Comparing these two regions, annual rainfall is higher in Campania than in Molise. In Sicily, the vineyards are on the eastern slopes of the Etna volcano. This region shows a highly seasonal rainfall pattern. The highest rainfall was registered during the first year; however, a long period of drought generally occurs during the period that corresponds to the phenological stages of late flowering, berry set, *vèraison*, and berry ripening. In the first year, despite a large amount of rain in August 2021, the subsequent dry period was very long (120 days). The pHs of the investigated soils were moderately alkaline (pH 8.3) at the Molise site, and neutral in Campania and Sicily (pH 7.1 and 7.0, respectively). All of the soils were classified as non-saline (Ec 0.135 dS$^{-1}$, 0.047 dS$^{-1}$, and 0.034 dS$^{-1}$ in

MOL, CAM, and SIC, respectively). The soil–water regimes during the growing season indicated differences in water availability for vines between the monitored sites (MOL > CAM > SIC), in accordance with the specific soil characteristics and climates (Table 1).

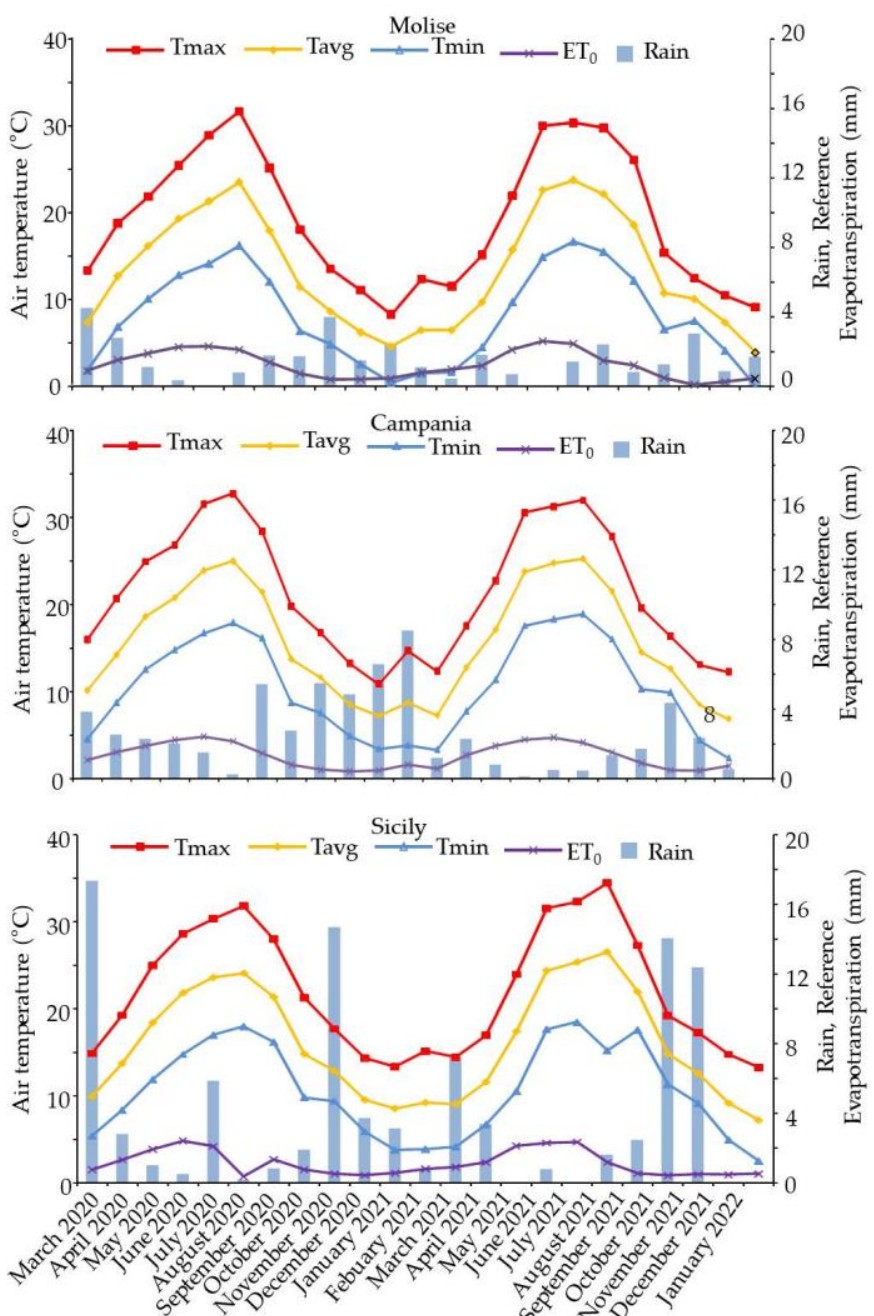

**Figure 1.** Monthly temperatures (°C; minimum—Tmin, average—Tavg, and maximum—Tmax), total rainfall (mm), and reference evapotranspiration (ET$_0$ mm) in the selected vineyards in the Molise (MOL), Campania (CAM), and Sicily (SIC) sites, during the years of experimental reporting.

### 3.2. Bud Fruitfulness

Data on bud fruitfulness are reported in Table 3. Shoot emergence from the main buds was always higher in AGL-CAM (2020) and AGL-MOL (2021), according to the training system adopted there. The *bourillon* and crown bud efficiencies, in terms of shoot emergence, were similar between sites; however, the highest values were observed for CAB, and particularly for CAB-MOL and CAB-CAM, in 2021. The main bunches produced by

AGL-MOL were about one per shoot in both years. For AGL-CAM in 2020, the number of bunches was low (average less than 0.5 per shoot). In the second year, 2021, a significant increase was observed that was in line with the retained main and lateral buds for each vine. The results in 2021 were similar to those of AGL-MOL in the same year. For CAB, the lowest shoot emergence was in Sicily, although the emerged bunches were higher than for AGL at the same site. The fertilities of the *bourillon* and crown buds were low at each of the three sites, but the highest values were observed in MOL in both years. For these parameters, the performance of CAB-SIC was similar to that of CAB-MOL. Data on blind buds revealed significantly higher values in CAM and MOL in 2020, and in MOL in 2021. CAB-SIC recorded the highest number of blind buds each year. The emergence of latent buds and their fertility in SIC were significantly lower in 2021 for AGL, while the lowest values for CAB were observed in MOL. The productivity, for AGL, was less than 1 in 2020, and less than 0.5 in 2021. Both latent shoots and their bunches were higher in CAB-SIC. CAB produced many shoots from latent buds that often showed a good degree of fertility.

**Table 3.** Vegetative and reproductive behaviours in each of two cultivars, Aglianico (AGL) and Cabernet Sauvignon (CAB), grown at three different sites, Molise (MOL), Campania (CAM), and Sicily (SIC). Measurements were made when all inflorescences were fully developed (Biologische Bundesanstalt, Bundessortenamt and CHemical industry: inflorescence 57). Mean values for each parameter, indicated by different letters, are significantly different (lowercase letter $p \leq 0.05$, uppercase letter $p \leq 0.001$, ± indicates one standard deviation), based on Tukey's HSD test within years and cultivars.

| Cultivar | Site | Main Shoots (*n*) | | Bourillon and Crown Shoots (*n*) | | Main Bunches (*n*) | | Bourillon and Crown Inflorescences (*n*) | |
|---|---|---|---|---|---|---|---|---|---|
| | | 2020 | 2021 | 2020 | 2021 | 2020 | 2021 | 2020 | 2021 |
| AGL | SIC | 2.53 ± 1.19 [C] | 3.33 ± 1.23 [C] | 1.53 ± 1.30 [c] | 2.40 ±0.74 [B] | 1.20 ± 0.86 [C] | 1.73 ± 0.59 [C] | 0.13 ± 0.09 [c] | 0.40 ±0.04 [C] |
| | CAM | 14.27 ± 1.91 [a] | 18.47 ± 1.64 [B] | 2.93 ± 1.28 [b] | 5.93 ± 1.39 [a] | 5.93 ± 4.32 [B] | 20.52 ± 2.61 [B] | 1.27 ± 1.10 [b] | 3.80 ± 1.37 [B] |
| | MOL | 12.53 ± 1.51 [b] | 22.93 ± 1.53 [A] | 4.73 ± 1.28 [a] | 6.53 ± 1.19 [a] | 12.13 ± 2.13 [A] | 26.40 ± 1.30 [A] | 2.20 ± 0.94 [a] | 12.07 ± 1.67 [A] |
| CAB | SIC | 5.87 ± 1.51 [B] | 6.67 ± 0.90 [B] | 3.13 ± 1.41 [B] | 2.33 ± 0.90 [C] | 10.13 ± 2.17 [B] | 8.33 ± 1.45 [C] | 3.67 ± 1.82 [a] | 1.13 ± 0.92 [C] |
| | CAM | | 19.27 ± 1.67 [a] | | 10.80 ± 1.42 [a] | | 21.60 ± 3.54 [a] | | 5.40 ± 1.40 [A] |
| | MOL | 12.27 ± 2.22 [A] | 18.60 ± 2.67 [a] | 5.53 ± 1.81 [A] | 9.07 ± 2.02 [b] | 13.07 ± 1.79 [A] | 19.13 ± 2.64 [b] | 3.00 ± 1.41 [a] | 3.13 ± 0.74 [B] |

| Cultivar | Site | Blind buds of main nodes (*n*) | | Main buds (*n*) | | Latent bud shoots (*n*) | | Latent bud bunches (*n*) | |
|---|---|---|---|---|---|---|---|---|---|
| | | 2020 | 2021 | 2020 | 2021 | 2020 | 2021 | 2020 | 2021 |
| AGL | SIC | 1.00 ± 0.53 [B] | 0.33 ± 0.04 [b] | 3.53 ± 1.25 [c] | 3.67 ± 1.29 [C] | 2.67 ± 0.98 [a] | 0.53 ± 0.02 [B] | 0.40 ± 0.83 [a] | 0.13 ± 0.06 [a] |
| | CAM | 2.47 ± 0.83 [a] | 1.00± 0.76 [a] | 16.76 ± 2.19 [a] | 19.47 ± 1.77 [B] | 2.13 ± 0.92 [a] | 5.73 ± 1.28 [a] | 0.80 ± 0.09 [a] | 0.33 ± 0.09 [a] |
| | MOL | 2.13 ± 0.99 [a] | 0.33 ± 0.05 [b] | 14.67 ± 2.16 [b] | 23.27 ± 1.49 [A] | 2.00 ± 0.76 [a] | 5.60 ± 0.99 [a] | 0.80 ± 0.68 [a] | 0.47 ± 0.09 [a] |
| CAB | SIC | 2.67 ± 1.59 [a] | 1.93 ± 0.96 [a] | 8.53 ± 1.92 [B] | 8.60± 1.40 [B] | 9.20 ± 3.12 [A] | 5.27 ± 1.16 [a] | 2.80 ± 1.27 [a] | 1.00 ± 0.76 [a] |
| | CAM | | 1.13 ± 0.92 [b] | | 20.40 ± 1.99 [a] | | 5.87 ± 1.41 [a] | | 0.73 ± 0.09 [ab] |
| | MOL | 0.93 ± 0.22 [b] | 0.73 ± 0.09 [b] | 13.20 ± 2.60 [A] | 19.33 ± 2.82 [a] | 4.73 ± 1.53 [B] | 0.60 ± 0.03 [B] | 0.67 ± 0.28 [b] | 0.20 ± 0.08 [b] |

In 2020, both the potential and observed bud fertilities were generally lower for AGL than for CAB, and the combination AGL-MOL had significantly higher values for both fertility indexes. In 2021, in MOL and in CAM, AGL confirmed this behaviour. For CAB, potential bud fertilities were higher in SIC, but there were no significant differences. For CAB in 2021, no differences were observed (Figure 2). Considering the vegetative and reproductive behaviours and the bud fruitfulness, in 2020, at each site, AGL showed the best correlations between main buds and main shoot emission (r = 92 in MOL, r = 92 in CAM, r = 90 in SIC), between observed bud fertility and main bunches (r = 0.81 in MOL, r = 0.96 in CAM, r = 0.85 in SIC), and between observed and potential bud fertility (r = 0.97 in MOL, r = 0.99 in CAM, r = 0.95 in SIC). All correlations that were detected in 2020 were confirmed in 2021, except that between the observed bud fertility and the main bunch, it was not significant in SIC. For CAB, the highest correlations were detected in MOL, between main buds and main shoot, between observed bud fertility and main buds, and between observed bud fertility and potential bud fertility, (r = 0.93, r = 0.80, r = 0.97, respectively).

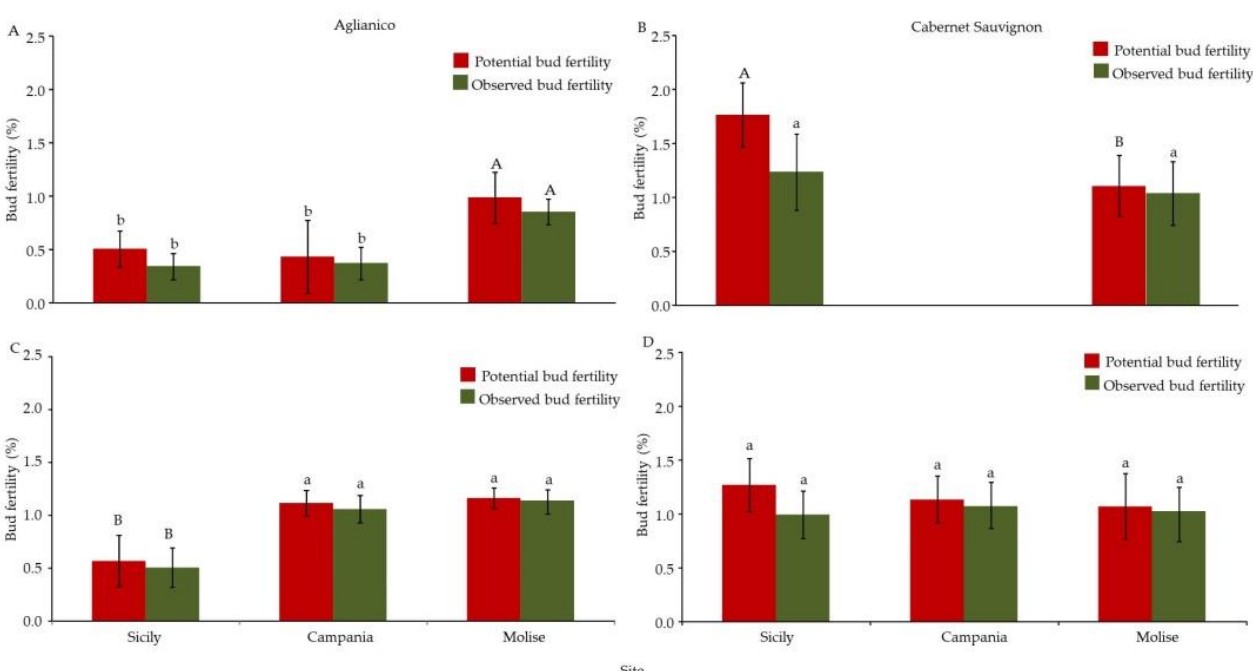

**Figure 2.** Potential and observed bud fertilities recorded in two cultivars, Aglianico (AGL) and Cabernet Sauvignon (CAB), grown in three different sites, Molise (MOL), Campania (CAM), and Sicily (SIC), over two years (**A**): Aglianico 2020; (**C**): Aglianico 2021; (**B**): Cabernet Sauvignon 2021; (**D**): Cabernet Sauvignon 2021). Measurements were made when all inflorescences were fully developed (Biologische Bundesanstalt, Bundessortenamt and CHemical industry: inflorescence 57). Mean values for each parameter, indicated by different letters, are significantly different (lowercase letter $p \leq 0.05$, uppercase letter $p \leq 0.001$, bars indicate one standard deviation), based on Tukey's HSD test within years and cultivars.

### 3.3. Morphological Response

The largest differences in the number of leaves were observed between the main and the lateral shoots, with the lateral shoots generally having fewer leaves. In 2020, the numbers of leaves along the main shoots of AGL were significantly fewer in SIC at the first observation (end of flowering), in MOL at the second observation (pea-sized berries), and MOL and CAM at the third observation (softening of berries) (Figure 3). For CAB, the main differences were in the number of leaves along the lateral shoots, among the sites. Meanwhile, in CAM and SIC, the number of leaves from lateral shoots were lower than the number of leaves from the main shoots; in MOL, at the berry softening stage, a strong growth of new leaves along the lateral shoots was observed. These results were due to the changing growth rates of the main and lateral shoots (not shown). Meanwhile, in SIC and CAM, from the mid-section of the shoot, the lateral emission was very limited (in several cases at least one incompletely developed leaf was observed); in MOL, the lateral growth continued during August (the hottest period of the summer), and the number of leaves along the lateral shoots increased.

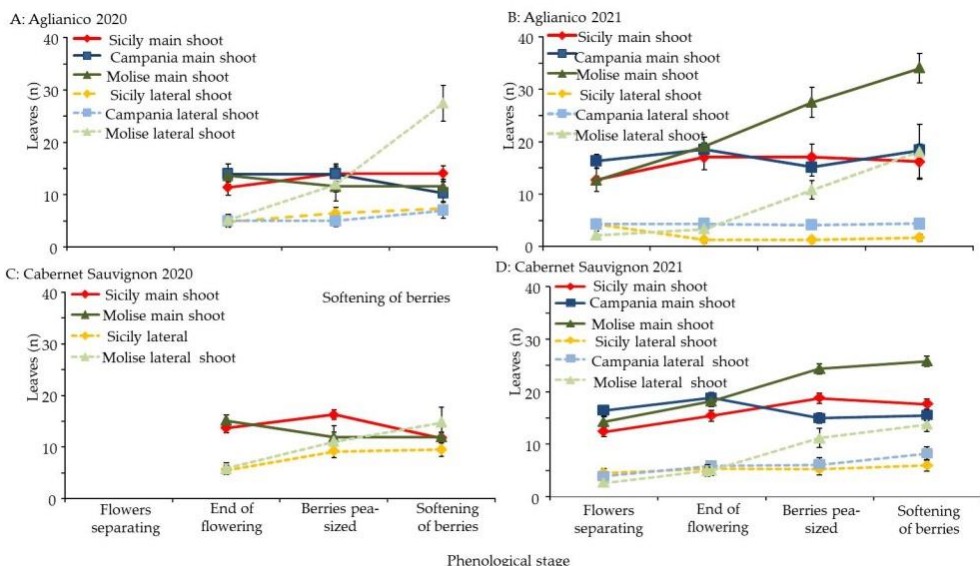

**Figure 3.** Numbers of leaves from main and lateral shoots recorded in two cultivars, Aglianico (AGL) and Cabernet Sauvignon (CAB), grown in three different sites, Molise (MOL), Campania (CAM), and Sicily (SIC), over two years (**A**): AGL 2020; (**B**): AGL 2021; (**C**): CAB 2020; (**D**): CAB 2021). Measurements were made at the phenological stages BBCH57 (only in the second year) flowers separating, BBCH69 end of flowering, BBCH75 berries pea-sized, and BBCH85 softening of berries (bars indicate one standard deviation), based on Tukey's HSD test within sites and years.

In both years, at each phenological stage, the leaf areas from the main shoots were always higher for AGL (Figure 4). In CAB, for the leaves on the lateral shoots in MOL and CAM at end of flowering in 2020, and in MOL at the pea-sized berries stage and at berry softening stage in 2021, the leaf areas were higher than those for the main shoots. In SIC, the lowest leaf area values were recorded for both the main and the lateral shoots.

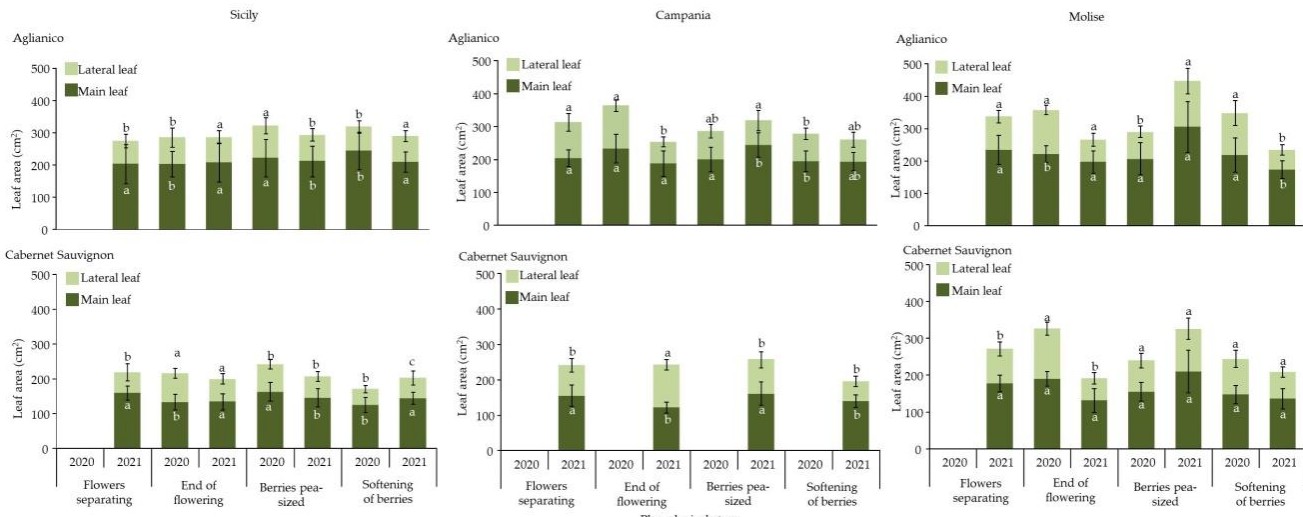

**Figure 4.** Leaf areas from the main and lateral shoots in each of two cultivars, Aglianico (AGL) and Cabernet Sauvignon (CAB), grown at three different sites, Molise (MOL), Campania (CAM), and Sicily (SIC). Measurements were made at the phenological stages BBCH57 (only second year) flowers separating, BBCH69 end of flowering, BBCH75 berries pea-sized, and BBCH85 softening of berries (bars indicate one standard deviation), based on Tukey's HSD test within sites and years.

Concerning the total leaf area per vine (Figure 5), AGL in MOL, during the first year at each phonological stage, showed significantly higher values. During the second year, except at the pea-sized berries stage, the vine behaviour was similar between CAM and MOL. In SIC, no differences were recorded between the different stages. For CAB, the trends for total leaf area per vine were similar to those for AGL.

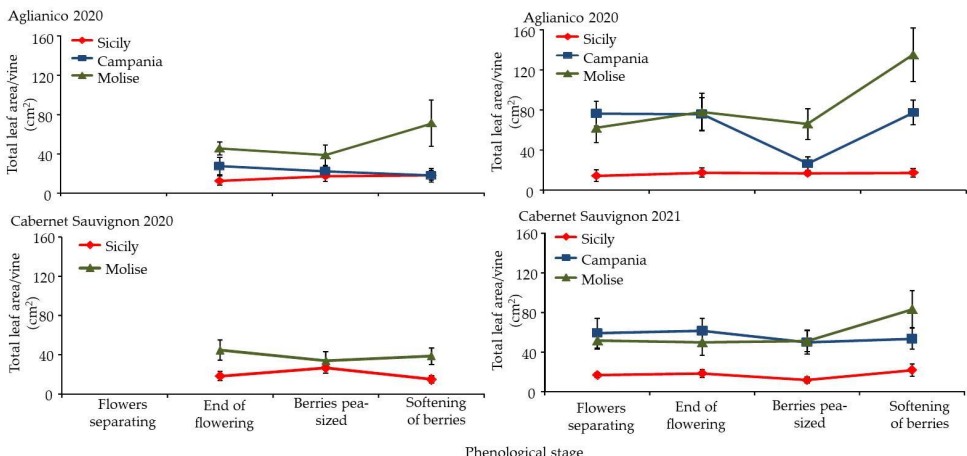

**Figure 5.** Total leaf areas per vine recorded in each of two cultivars, Aglianico (AGL) and Cabernet Sauvignon (CAB), grown in three different sites, Molise (MOL), Campania (CAM), and Sicily (SIC). Measurements were made at the phenological stages BBCH57 (only second year) flowers separating, BBCH69 end of flowering, BBCH75 berries pea-sized, and BBCH85 softening of berries (bars indicate one standard deviation), based on Tukey's HSD test within sites and years.

The leaf area indexes for AGL and CAB during the first year are reported in Figure 6. A significant difference was recorded for AGL at the pea-sized berries stage, in which the highest value was observed in SIC. Conversely, at the berry softening stage, the leaf area index of AGL-MOL showed a significant increase. This trend was especially evident in 2021. CAB showed significant differences between SIC and MOL only at the pea-sized berries stage.

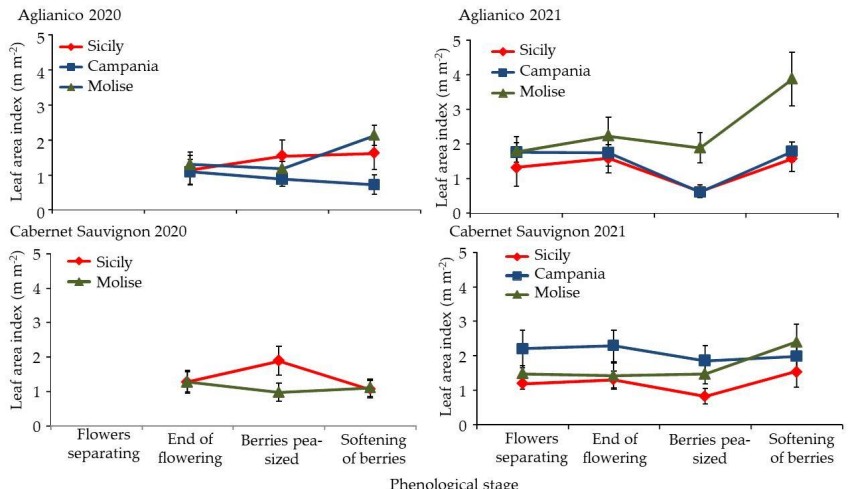

**Figure 6.** Leaf area indexes registered in each of two cultivars, Aglianico (AGL) and Cabernet Sauvignon (CAB), grown at three different sites, Molise (MOL), Campania (CAM), and Sicily (SIC). Measurements were made the phenological stages BBCH57 (only second year) flowers separating, BBCH69 end of flowering, BBCH75 berries pea-sized, and BBCH85 softening of berries, during the second year. Mean values for each parameter indicated by different letters are significantly different ($p \leq 0.001$) (bars indicate one standard deviation), based on Tukey's HSD test within sites and years.

### 3.4. Vines Physiological Behaviour

The leaf water potential for AGL in 2020 showed differences at the end of flowering and berry softening stage (Figure 7). At the berry softening stage, a significantly lower negative value was recorded for the lateral leaves in CAM. In the second year, AGL showed the highest negative water potential values in SIC up to the pea-sized berry stage, both for the main and lateral leaves. At the second and third stages in CAM, less negative values were observed. For CAB in 2020, the less negative water potential was recorded in SIC at the end of flowering and at the pea-sized berries stages, both for main and lateral shoots. In 2021, a different behaviour was observed between SIC and MOL during flower separating and the end of flowering.

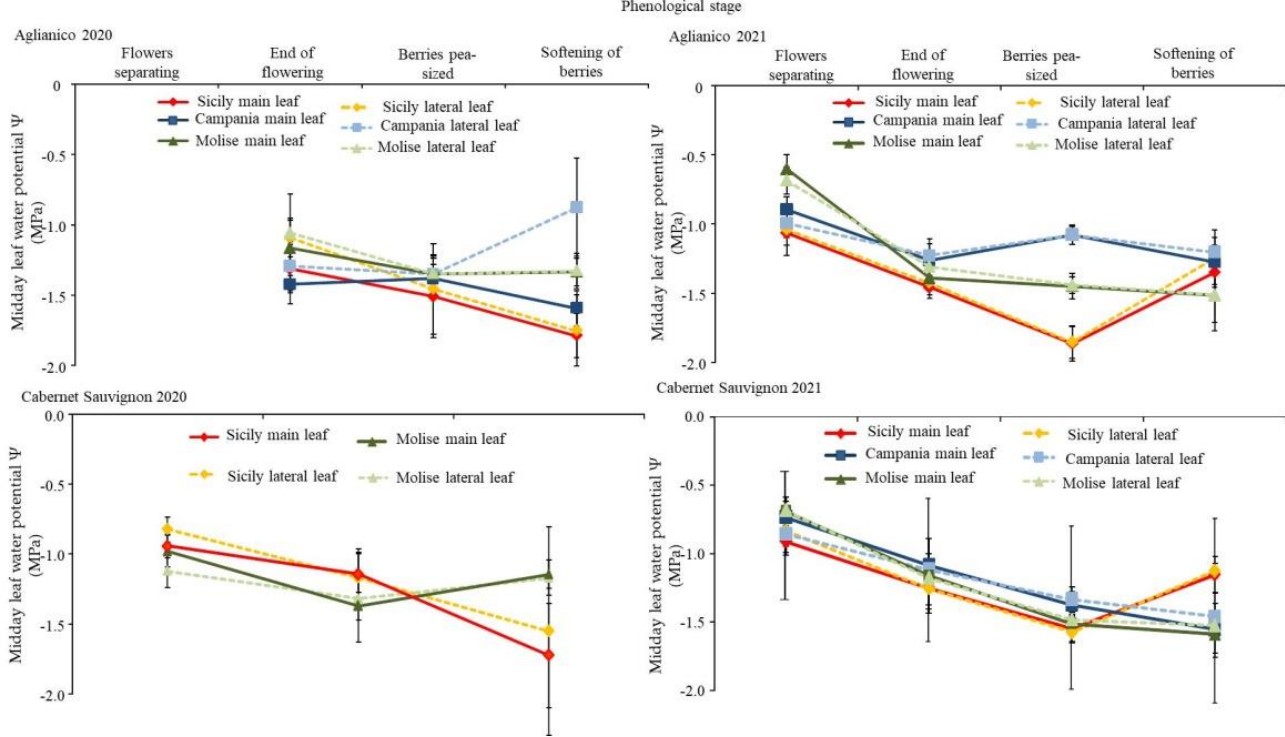

**Figure 7.** Midday leaf water potentials measured on main and lateral leaves recorded in each of two cultivars, Aglianico (AGL) and Cabernet Sauvignon (CAB), grown in three different sites, Molise (MOL), Campania (CAM), and Sicily (SIC). Measurements were made at the phenological stages BBCH57 (only second year) flowers separating, BBCH69 end of flowering, BBCH75 berries pea-sized, and BBCH85 softening of berries (bars indicate one standard deviation), based on Tukey's HSD test within sites and years.

### 3.5. Crop Yield and Characteristics

The reproductive characteristics are reported in Table 4. AGL had the highest yield and bunch weight in MOL, in both years. No differences were found for bunch length and berry number, while the berry and rachis weights were significantly lower in SIC in 2020 and 2021. CAB showed a higher yield in MOL in 2021.

**Table 4.** Yield and its components in each of two cultivars, Aglianico (AGL) and Cabernet Sauvignon (CAB), grown in three different sites, Molise (MOL), Campania (CAM), and Sicily (SIC). Measurements were made when all inflorescences were fully developed (Biologische Bundesanstalt, Bundessortenamt and Chemical industry: inflorescence 57). Mean values for each parameter and years indicated by different letters are significantly different (lowercase letter $p \leq 0.05$, uppercase letter $p \leq 0.001$, n.s. not significant $\pm$ indicates one standard deviation), based on Tukey's HSD test within years and cultivars.

| | AGL | | | | | |
|---|---|---|---|---|---|---|
| **Parameters** | **MOL** | | **CAM** | | **SIC** | |
| | **2020** | **2021** | **2020** | **2021** | **2020** | **2021** |
| Yield (t/ha) | 19.53 ± 5.39 [A] | 26.16 ± 3.76 [a] | 4.86 ± 1.63 [B] | 14.34 ± 5.11 [b] | 2.01 ± 0.89 [B] | 4.09 ± 0.87 [b] |
| Bunch weight (g) | 558.60 ± 161.5 [A] | 345.80 ± 42.08 [n.s.] | 348 ± 56.35 [b] | 298.20 ± 104.28 [n.s.] | 190 ± 15.41 [B] | 315.40 ± 85.23 [n.s.] |
| Bunch length (cm) | 16.80 ± 0.84 [n.s.] | 19.40 ± 1.82 [n.s.] | 16.80 ± 2.49 [n.s.] | 19.20 ± 1.10 [n.s.] | 16 ± 1.87 [n.s.] | 18.20 ± 1.30 [n.s.] |
| Berry number | 147.88 ± 88.59 [n.s.] | 126.50 ± 13.10 [n.s.] | 129.96 ± 25.11 [n.s.] | 129.80 ± 51.69 [n.s.] | 79.80 ± 7.66 [n.s.] | 106.6 ± 24.47 [n.s.] |
| Berry weight (g) | 2.64 ± 0.22 [a] | 2.59 ± 0.22 [b] | 2.58 ± 0.17 [a] | 2.19 ± 0.27 [a] | 2.15 ± 0.03 [b] | 2.38 ± 0.19 [ab] |
| Rachis weight (g) | 16.80 ± 1.64 [a] | 18.80 ± 2.77 [a] | 14.60 ± 2.07 [ab] | 17.40 ± 1.14 [ab] | 18.46 ± 1.50 [b] | 14.44 ± 1.24 [b] |

| | CAB | | | | | |
|---|---|---|---|---|---|---|
| | **MOL** | | CAM | | SIC | |
| | **2020** | 2021 | 2020 | 2021 | 2020 | 2021 |
| Yield (t/ha) | 13.91 ± 2.74 [A] | 24.61 ± 4.47 [A] | - | 18.36 ± 8.56 [b] | 17.15 ± 2.97 [B] | 19.50 ±4.58 [B] |
| Bunch weight (g) | 352.60 ± 74.82 [a] | 444.6 ± 64.92 [n.s.] | - | 318.80 ± 102.27 [n.s.] | 258.40 ± 25.38 [b] | 346.60 ± 56.68 [n.s.] |
| Bunch length (cm) | 20.60 ± 2.07 [b] | 18.60 ± 1.95 [n.s.] | - | 19.20 ± 1.10 [n.s.] | 17.60 ± 1.52 [a] | 19.00 ± 1.87 [n.s.] |
| Berry number | 169.20 ± 35.97 [a] | 154.00 ± 15.07 [n.s.] | - | 161.40 ± 69.09 [n.s.] | 125.80 ± 19.23 [b] | 238.85 ± 74.35 [n.s.] |
| Berry weight (g) | 1.97 ± 0.17 [n.s.] | 2.75 ± 0.18 [A] | - | 1.89 ± 0.22 [b] | 1.90 ± 0.12 [n.s.] | 1.50 ± 0.23 [c] |
| Rachis weight (g) | 18.60 ±1.82 [b] | 19.00 ± 2.35 [n.s.] | - | 20.60 ±3.05 [n.s.] | 21.20± 1.64 [a] | 17.40 ±1.82 [n.s.] |

### 3.6. Vine Behaviour

A PCA was carried out in order to evaluate the effects of site on vine development (Figure 8A–D). A clear separation was observed between MOL and SIC in both years. In SIC, the parameters appear in the negative quadrant of component 2 as a consequence of the highly negative water potential. The morphological results for SIC are lower than in MOL or CAM. A similar trend was detected in MOL for midday leaf water potential of CAB at the pea-sized berry stage, and of AGL at the flower separation stage. For AGL (Figure 8B), component 1 accounted for 41.73% of the explained variance, while component 2 accounted for 21.02% of the explained variance. This shows that AGL-SIC and AGL-CAM follow a similar trend in the negative quadrant of component 1 as a result of the negative water potential. In MOL, except at the flower separation stage in 2021, all the parameters separate into the positive quadrant for component 1. For CAB (Figure 8B,D), component 1 accounted for 37.79% of the explained variance, while component 2 accounted for 22.45% of the explained variance.

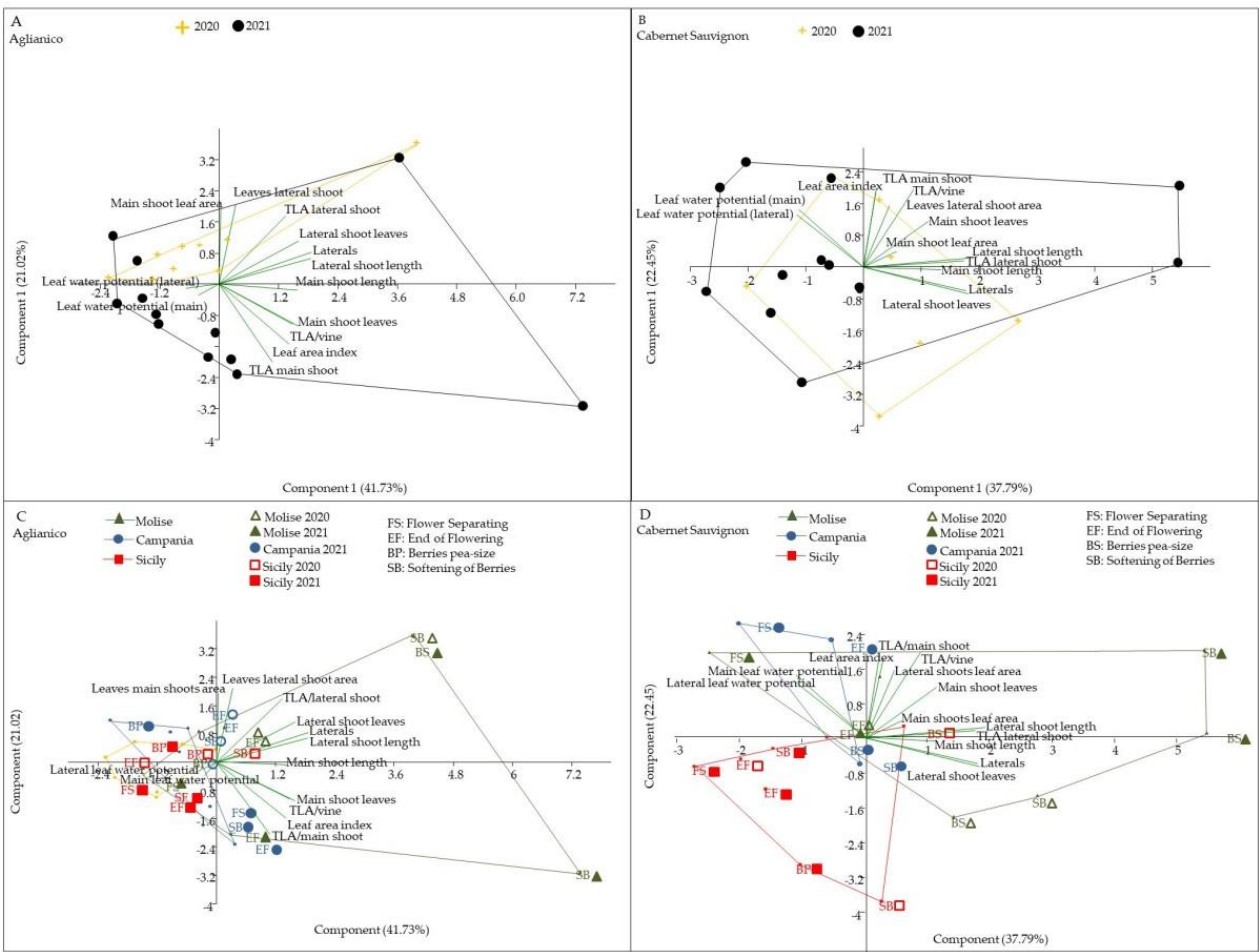

**Figure 8.** Biplot resulting from principal component analysis of the average data for Aglianico (AGL) and Cabernet Sauvignon (CAB), over two years (2020–2021), from three regions, Molise (MOL), Campania (CAM), and Sicily (SIC) in southern Italy, as defined by the first two principal components. The cultivars are clustered on the basis of morphological indices and midday leaf water potential at each phenological stage. The two cultivars are separated by year (**A**) for AGL and (**B**) for CAB), and by region (**C**) for Aglianico and (**D**) for Cabernet Sauvignon). Groups are delimited by convex hulls to highlight their respective trends.

## 4. Discussion

This study explored the interaction genotype x environment for two black grape cultivars, AGL and CAB. The first is a traditional wine grape cultivar of central and southern Italy, while Cabernet Sauvignon is a French cultivar that is widely grown in the main viticultural areas of the world, including Italy. For these cultivars, the thermal needs expressed by [54] were found to be about 2119 growing degree days (GDD) (threshold temperature 10 °C) for AGL [55], and about 1900 GDD for CAB [56]. Considering the two years of weather data, the three experimental sites differ from each other in terms of annual temperatures: the coldest site is MOL, and the hottest is SIC. During summer, the weather conditions are similar for daily mean temperatures in CAM and SIC. In detail, the main characteristic of the weather in SIC is the long dry period that begins in late spring and continues to the end of summer, with extreme maximum temperatures occurring during mid-summer. In SIC, the presence of a shallow soil combined with a warm climate produce a soil water deficit that has direct effects on fruit yield and berry composition [57]. On the contrary, the water availability for the vine in MOL is higher compared to the other sites, thanks to the specific weather conditions and soil characteristics (deep, well-structured

clay soil without rooting impediments), leading to a lower plant water stress. In CAM, the specific pedoclimatic conditions (deep clay loam soil without rooting impediments) produce intermediate environmental conditions that are between SIC and MOL in the growing season for the vine.

The results presented here offer a measure of phenotypic plasticity under the conditions of this study. That is, they relate to the regions, species, and cultivars considered here, and are unlikely to be simply comparable with studies where the *terroirs*, species, and cultivars are different [58].

This study considered both vegetative and reproductive bud development over a two-year period, as affected by the environmental component of *terroir*—i.e., different climate, soil, and cultural conditions. Bud fruitfulness can be assessed on the basis of the parameters associated with the blind bud. Particularly in the first year, in CAM and MOL, blind bud fruitfulness was very low compared to the retained main bud. On the contrary, in SIC mainly during the first year, 28% and 31% of blind buds were observed for AGL and CAB, respectively. The blind bud value decreased to 22% and 9%, respectively, during the second season. Considering the lower bud load of the vineyards in SIC, these percentages could be considered slightly high. CAB-SIC, emitted the highest number of shoots from latent buds and some of these flowered. In order to analyse bud fruitfulness, it is necessary to consider not only the weather and soil conditions of a particular season, but also those at the time of flowering in the previous year, and particularly the water and N stresses at that time [59,60]. Guilpard et al. [43] reported that decreases in leaf nitrogen and moderate water stress at flowering in the previous season can reduce bud fertility in the subsequent season by 36% to 40%, although this response is cultivar dependent.

The vegetative parameters indicate a long period of water stress [61]. Among the morphological parameters, the number of leaves recorded during the first stage highlighted a surprising late growth in Sicily. The main differences among sites and cultivars were the time interval between the end of flowering (fruit set), and the softening of berries (*vèraison*). The main driver of growth and development during this period that determined the length of this phenological interval was the temperature. It is possible that rapid soil drying and extreme temperatures cause early root death in shallow soil, hence having a greater effect on the vine compared to the deeper soils [62]. This would reduce the growth and emergence of new leaves. This behaviour was different in AGL, which showed a reduction in leaf number at the end of flowering and at the pea-sized berries stages in CAM and MOL; however, leaf number was constant in SIC in 2020. The reduction in leaf number was due to hedging, which was required to combat a late season *Plasmopara viticola* infection (data not shown). In 2021, the differences in leaf numbers were due to the high water stress in AGL-SIC. During this summer period, a drought induced wilting and death of basal leaves, with the main shoots observed to retain only the mid and apical leaves. However, in CAM and mainly in MOL, the shoots grew continuously, and new leaves continued to emerge. This latter behaviour was not observed in CAB-SIC.

Leaf area is highly influenced by a cultivar's genetic characteristics. It is well known that AGL normally has large leaves, while CAB leaves are generally smaller. The observed differences in leaf area among regions were likely due to terroir (climate + soil + management). From mid-summer until the end of the growing season, the lateral leaf growth in AGL was not much influenced by the terroir conditions in CAM and MOL; meanwhile, it was highly influenced in SIC. In SIC, a much-lowered emergence of lateral shoots in AGL from the pea-sized berries stage to the *vèraison* stage was observed. In several cases, only one incompletely expanded leaf emerged from lateral shoots in the mid trait of the cane, while no leaves emerged on the lateral shoots from the upper nodes of the canes.

The climate, the soil conditions, and the management were responsible for the growth rate reductions in total leaf area per vine in AGL, from the pea-sized berries stage to the *vèraison* stage in CAM and SIC. For CAB, the best performance in terms of leaf area was observed in CAM, between the flower separating and fruit set stages, with lower 'performances' in MOL and SIC. At the pea-sized berries stage in MOL and CAM, CAB

showed similar behaviours that were significantly higher than those in SIC. The reductions in leaf area index that were observed in mid-summer in CAM and in MOL were due to hedging of the vines that occurred during that period. The leaf area index of AGL-SIC in 2020 was not determined by any particularly limiting weather and soil conditions. The long growth period in MOL produced the highest leaf area index, late in the season. Although vines were pruned differently, and the bud loads were different among sites, the differences in TLA and LAI can probably be ascribed to the depressive effects of drought on shoot elongation and leaf emergence [63]. Vine aboveground fresh mass is directly related to the belowground fresh mass (i.e., a large canopy corresponds to a large root system, and *vice versa*) [18]. Therefore, although is possible to hypothesize that the soil in SIC offered low resistance to root elongation, the smaller canopies in AGL-SIC may be ascribed to a lower availability of water, according to the soil physical characteristics. During early summer, midday leaf water potential was about $-0.8$ MPa, and as low as $-1.0$ MPa; from mid-summer to the end of the season, even more negative values were recorded. Considering that for well-irrigated vines, midday leaf water should be approximately $-0.8$ MPa [23], our conditions were similar to those of an imposed drought from the fruit set through to the *vèraison* stage. During this period, values as low as $-1.2$ MPa or even $-1.5$ (CAB) were recorded. These should be considered as moderately stressed or severely stressed conditions. For AGL-SIC, the most negative values reached about $-2.0$ MPa. This lower (more negative) leaf water potential could be related to the larger leaf area that increases canopy water use. Meanwhile, in CAB, a reduction in leaf area should result in a less negative water potential [22].

The CAB response was driven mainly from its well-known vegetative habits (compact canopy, small leaves). The low values of leaf number and shoot growth were due to the high plasticity of this cultivar that adapted its phenotype to the limiting growing conditions, such as those recorded in SIC in 2021. The positions of AGL-SIC and AGL-CAM into the negative quadrant are a consequence of negative leaf water potentials in both the main and lateral shoots (Figure 8). Despite the negative trend for water potential that was found also for CAB at *vèraison*, and for AGL-MOL at flower separation, the other parameters were generally higher for AGL at *vèraison* and at the end of the growing season for TLA/main shoot, leaf area index, TLA/vine, main shoot leaf area, and lateral shoot leaf area. These results confirm the long vegetative and reproductive cycles of AGL mainly in colder areas, and hence, the high canopy efficiency (continuous growth) at the end of summer during a hot season, such as that in 2021. Between the two years, the PCA analysis revealed a greater difference under increasing summer temperatures from MOL to SIC in 2021. The production was very low for AGL, probably below the economic convenience of the grower. A higher bud load would probably be better for AGL production in Sicily. However, the reduced production, mainly due to low bunch weight in a vine that is usually highly productive, such as in AGL-MOL and AGL-CAM, is interesting with regard to a standard that aligns with the requirements of a quality viticulture, especially in a protected area of origin wine-producing area. In a vigorous cultivar such as CAB, in which the bunch is generally small and short, the yield reduction was excessive compared to its potential. It is well established that the main drivers of grapevine yield are bunch number per vine, and berry number per bunch, which account for about 60% and 30% of seasonal yield variation, respectively. Meanwhile, berry weight variation accounts for only about 10% of seasonal yield variation [64,65], with the flowering and green berry stages critical to the determination of berry size [66]. Either excessive soil water availability or severe drought can reduce vineyard yield, with excessive water leading to luxuriant vegetative growth at the expense of reproductive growth [63]. The first and the second cases probably occurred in MOL and SIC, respectively. Therefore, for AGL, G x E interactions seemed to have their main effects during berry ripening, while for CAB, G x E plasticity seemed to be more limited, perhaps explaining why CAB is so widely successful [4]. In general, the vegetative and productive responses of the two vines in the three environments highlight how the vegetative and productive results, and therefore the different parameters of the

CAB, yielded traits that were less dependent on the different environmental conditions, remaining more uniform between the regions, and confirming the limited plasticity of this cultivar. On the contrary, AGL was found to be highly dependent on climatic conditions. In particular, in SIC, the hottest region, limited vegetative growth was observed. This behavior was due to a low bud fruitfulness, a limited shoot elongation, a limited lateral growth, and also to the early leaf drop in the basal part of the shoots. An opposite behavior was observed in MOL, where the climatic conditions led to an excess of vegetative growth and, perhaps, an excessively high productive level. Furthermore, the AGL response was highly variable over the two years, and within the growing season of the year. Figure 9 shows some bunches of the two vines in the three different sites during the ripening phase.

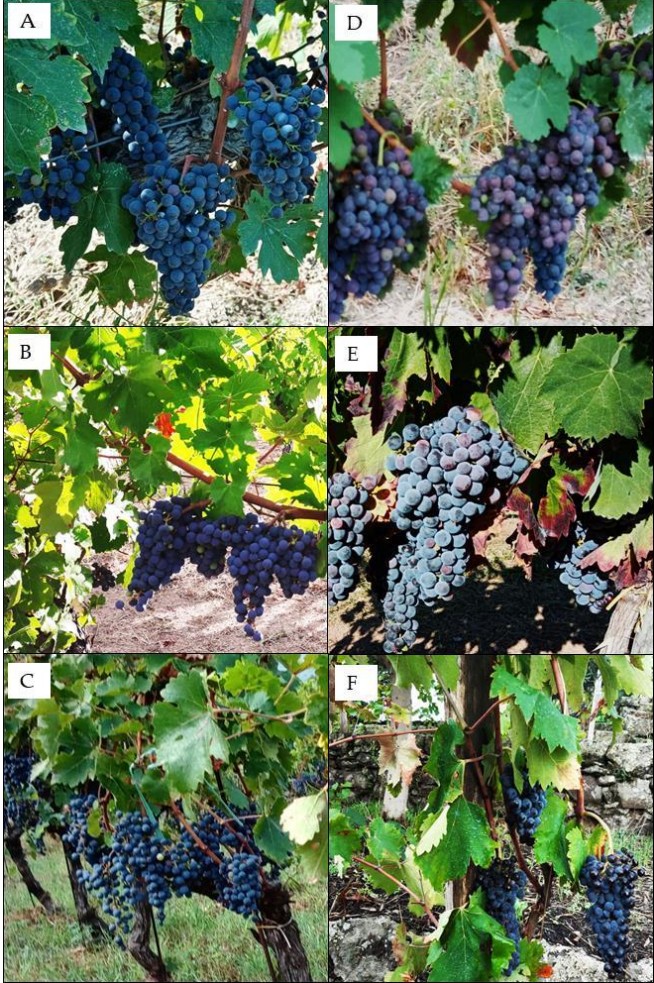

**Figure 9.** Bunches of the two cultivars, Cabernet Sauvignon (CAB) and Aglianico (AGL), grown in three different sites, Molise (MOL), Campania (CAM), and Sicily (SIC). (**A**): CAB MOL; (**B**): CAB CAM; (**C**): CAB SIC; (**D**): AGL MOL; (**E**): AGL CAM; (**F**): AGL SIC.

## 5. Conclusions

Bud fruitfulness, across the three locations in SIC, MOL, and CAM, was more variable for AGL than for CAB; however, vine growth was also influenced by the different pruning systems used in the three sites. The leaf water potential of AGL was very negative during the dry season, especially in 2021. Production was in line with the bud load chosen during winter pruning. A high phenotypic plasticity was observed for AGL: the plasticity increased as a result of limiting environmental conditions, and also compared to the environmental conditions that favoured an excess of vegetative growth (e.g., mild temperatures and rains at different phenological stages); this contributed to excessive growth and canopy

density. On the contrary, CAB expressed its morphological stability mainly through its bud performance, shoot growth, leaf area, and total leaf area/vine. Meanwhile, AGL performed best, and showed the best vegetative/reproductive balance in CAM, which is also its area of origin. The stability of CAB confirms the possibility of success of this cultivar in different locations, or in a scenario involving climate change. The reported results could be useful both to growers, in order to better understand which cultivar could be more suitable for different environmental conditions. The results are also of interest to ecologists for studies on the effects of climate change on vine performance.

**Author Contributions:** Conceptualization, F.F., A.S., E.N. and A.R.L.P.; methodology, F.F., A.S. and E.N.; validation, E.N., F.F., A.S. and A.R.L.P.; investigation, E.N., F.F., A.B., E.M. and A.S.; data curation, F.F., E.M. and E.N.; writing—original draft preparation, A.S., F.F. and E.N.; writing—review and editing, E.N., F.F., A.B., E.M. and A.S.; supervision, F.F., E.N., A.S. and A.R.L.P.; funding, A.R.L.P. All authors have read and agreed to the published version of the manuscript.

**Funding:** Progetti di ricerca di Rilevante Interesse Nazionale (PRIN2017) "Influence of Agro-climatic conditions on the microbiome and genetic expression of grapevines for the production of red wines: a multidisciplinary approach (ADAPT)", Ministero dell'Università e della Ricerca (MUR).

**Acknowledgments:** The authors would like to thank the farms Scammacca del Murgo and Pecora (Sicily), Porto di Mola (Campania), and Giagnacovo (Molise), for hosting the trials.

**Conflicts of Interest:** The authors declare no conflict of interest.

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
