# Peer review of "Phenotypic Plasticity in Bud Fruitfulness Expressed in Two Distinct Wine Grape Cultivars Grown under Three Different Pedoclimatic Conditions"

_agriculture, doi:10.3390/agriculture12101660_

Round 1
Reviewer 1 Report (Previous Reviewer 1)
The stability of a variety mainly depends on the size of the interaction effect between genotype and environment. Therefore, the objective evaluation of the traits of a variety requires the investigation of the variety in multiple sites over many years.this research is systematic and comprehensive, It is suggested that the paper be accepted after minor modification, the main suggestions are as follows:
1.In table 3, why the p-value is chosen as 0.001 instead of 0.01?
2. For a certain variety, some characters (such as main shoots, main bunches) showed significant differences at different sites. What is the reason, is it just due to the climate? Are there any other environmental factors that suggest the author to conduct systematic hand binding according to the test results.
Author Response
Reviewer 1
The stability of a variety mainly depends on the size of the interaction effect between genotype and environment. Therefore, the objective evaluation of the traits of a variety requires the investigation of the variety in multiple sites over many years. this research is systematic and comprehensive, it is suggested that the paper be accepted after minor modification, the main suggestions are as follows:
Thank you so much for your great comment.
- In table 3, why the p-value is chosen as 0.001 instead of 0.01?
In table 3 the choice of the p-value is in agreement with the results of ANOVA analysis. In particular, for some parameters the data procession showed the Fisher’s F values less than 0.001, therefore in according the mean separation by the Tukey’s test was performed at this probability level.
- For a certain variety, some characters (such as main shoots, main bunches) showed significant differences at different sites. What is the reason, is it just due to the climate? Are there any other environmental factors that suggest the author to conduct systematic hand binding according to the test results?
Thank you for the comment. As reported in m&m, section 2.1. Site descriptions, plant material and trial design, the training systems and the bud load were different mainly in Sicily. Therefore, as far as the mentioned parameters (main shoots and bunches) the statistical significance derived from the cited difference. Of course the environmental conditions improved the blind bud level in that region and this was reported in the text. The studied variables, as reported into the text, are related to a number of distinct growth processes such us floral induction, organogenesis, biomass production and partitioning, and the considered climatic variables were responsible for the detected differences among sites and cultivars. In our opinion a strong role was due to the soil characteristics and to the soil conditions mainly during the summer period. On the contrary the age of the vines and the common rootstock that were similar were not responsible for the significant differences. All these variables effects were hypothesized along the manuscript.

Reviewer 2 Report (New Reviewer)
I have attached a main text which contains all comments and questions.

Author Response
Reviewer 2
I have attached a main text which contains all comments and questions:
- Lines 82-83: Among the environmental stresses that can affect growth are: drought, high temperatures, and excess solar radiation. Only these? How about other abiotic stress? Heavy metals, UV and etc... So it is better to say some stress not all
The sentence was changed as follows: Among the main environmental stresses that can affect growth are: drought, high temperatures, excess solar radiation, ultraviolet radiation, heavy metals, salts.
- Line 94: What do you mean strongest and rapid response?
The sentence was changed as follows:
In general, the earliest and high evident response to a new environmental limitation is a change in plant vigour.
- The results were really interesting, as the authors correctly addressed one of the main problems in grapevine production. Similar name for different cultivars in different regions and different name for same cultivars too. So this type of report could be very useful in this regards. The experiment was designed well. But I strongly recommend to bring some photo from leaf and berry in different location as the main goal of this experiment was the phenotyping.
Thank you very much for your comment. We try to improve the paper adding a photo of leaves and bunches from different sites (Figure 9 line 596)
- - s solar radiation. Only these Line 276 in Molise site and neutral in Campania and Sicily (extra space)
Revised
- Table 3
What is the reason for increasing shoot number in second year?
As reported in M&M the bud load was similar between the years. The reduction in main shoots was due to a reduction in blind bud.

Reviewer 3 Report (New Reviewer)
The MS “Phenotypic plasticity in bud fruitfulness expressed in two distinct wine-grape cultivars grown under three different pedo-climatic conditions” is well-structured with all supportive data in relation to claims made. In this ms, two wine-grape cultivars (native Aglianico and international Cabernet Sauvignon) were subjected to different pedo-climatic conditions. The MS is scientifically sound and has very important information to the scientific community and growers.
Author Response
Reviewer 3
The MS “Phenotypic plasticity in bud fruitfulness expressed in two distinct wine-grape cultivars grown under three different pedo-climatic conditions” is well-structured with all supportive data in relation to claims made. In this ms, two wine-grape cultivars (native Aglianico and international Cabernet Sauvignon) were subjected to different pedo-climatic conditions. The MS is scientifically sound and has very important information to the scientific community and growers.
Thank you so much for your great comment.

This manuscript is a resubmission of an earlier submission. The following is a list of the peer review reports and author responses from that submission.
Round 1
Reviewer 1 Report
Plant phenotypic plasticity is the different phenotypic characteristics of the same genotype affected by different environments. It is a trade-off between plant traits and heterogeneous environments, and is an adaptation and expression of plants to the environment. At present, it has become an important field of ecological research,especially in the context of climate change, the response and adaptation of plant phenotypic plasticity to climate change is of great significance. The results of the study are very interesting, and the relevant review recommendations are as follows:
(1) How relevant are the relevant phenotypic data for the two years, please explain in the discussion in conjunction with environmental and climate data.
(2) In table 3, Different locations have different grapevine densities. It is recommended that the yield data be uniformly converted to yield per hectare.
(3) Mineral nutrition also has a great influence on the flower bud differentiation of grapes. It is suggested to provide supplementary explanation whether the soil nutrient data of 3 vineyards have been collected.
(4) In line 105, please change 36 to 36%.
(5)The format of some references is not standardized, please check and modify.
Reviewer 2 Report
In general, the degree of innovation of this article is very low and it does not add new and useful knowledge to viticulture.
The main problems are summarized as follows:
- The title of paper is “Phenotypic plasticity in bud fruitfulness expressed in two distinct winegrape cultivars grown under three different pedoclimatic conditions”, but the pedoclimatic conditions are not descripted. No pedoclimatic parameters as - just to give some examples - soil temperature and moisture, have been monitored in this study. The soil and pedoclimatic parameters homegenity of plot within the vineyards have not been analised or described.
- The canopy management, in particular the shoot topping, is not described.
- The surveys were carried out only for two years, and the climatic conditions were very different in 2020 and 2021, moreover, some parameters have been measured once. At least, tree years of surveys are required for similar studies.
- The citation numbers are sometimes malplaced or the references are wrong. In general all “References” have to be corrected. Just to give some examples:
o Reference [16] - is not correct, the cited paper is not about Cabernet sauvignon and Syrah.
o Reference [40] - you can find papers about grapevine
o Line 584 - the autor is Luca Brillante, not A.
o Reference [18] – is wrong, the correct one is: Scholander, P.F.; Hammel, H.T.; Bradstreet, E.D.; Hemmingsen, E.A. Sap pressure in vascular plants. Science 1965, 148, 339–346.
In short, this study is useless since it does not reach an adequate conclusion: therefore the manuscript is not suitable for publication.
Reviewer 3 Report
The manuscript is very well written, but still some comments.
Please pay attention to the length of each mart of the manuscript. E.g. The abstract should be no more than 200 words. It is more than 300 words at the moment.
As the manuscript ought to provide some new insight into the connections between genotype and enviroment and show the plasticity of the different grapevine cultivars. Please conclude more clearly if there were found any new inshights of the complex system that connects genotype and environment to deliver phenotype or not, as promised in the introduction part.
Also please provide the information, to whom this research result is useful and relevant in the light of your research results.